# The Role of the Cumulative Illness Rating Scale (CIRS) in Estimating the Impact of Comorbidities on Chronic Obstructive Pulmonary Disease (COPD) Outcomes: A Pilot Study of the MACH (Multidimensional Approach for COPD and High Complexity) Study

**DOI:** 10.3390/jpm13121674

**Published:** 2023-11-30

**Authors:** Domenico Di Raimondo, Edoardo Pirera, Chiara Pintus, Riccardo De Rosa, Martina Profita, Gaia Musiari, Gherardo Siscaro, Antonino Tuttolomondo

**Affiliations:** 1Division of Internal Medicine and Stroke Care, Department of Promoting Health, Maternal-Infant, Excellence and Internal and Specialized Medicine (ProMISE) “G. D’Alessandro”, University of Palermo, 90133 Palermo, Italy; edoardo.pirera@unipa.it (E.P.); chiarapintus1809@gmail.com (C.P.); derosariccardo96@gmail.com (R.D.R.); martinaprofita9@gmail.com (M.P.); gaiamusiari@gmail.com (G.M.); bruno.tuttolomondo@unipa.it (A.T.); 2Medical Affairs, Chiesi Italy SpA, 43122 Parma, Italy; g.siscaro@chiesi.com

**Keywords:** comorbidity, multimorbidity, chronic obstructive pulmonary disease (COPD), Cumulative Illness Rating Scale, acute exacerbation of COPD (AECOPD)

## Abstract

Background. Chronic obstructive pulmonary disease (COPD) is a heterogeneous systemic syndrome that often coexists with multiple comorbidities. In highly complex COPD patients, the role of the Cumulative Illness Rating Scale (CIRS) as a risk predictor of COPD exacerbation is not known. Objective. The objective of this study was determine the effectiveness of the CIRS score in detecting the association of comorbidities and disease severity with the risk of acute exacerbations in COPD patients. Methods. In total, 105 adults with COPD (mean age 72.1 ± 9.0 years) were included in this prospective study. All participants at baseline had at least two moderate exacerbations or one leading to hospitalization. The primary outcome was a composite of moderate or severe COPD exacerbation during the 12 months of follow-up. Results. The CIRS indices (CIRS total score, Severity Index and Comorbidity Index) showed a positive correlation with modified Medical Research Council (*mMRC*), COPD assessment test (CAT) and a negative correlation with forced expiratory volume in the first second (FEV_1_), Forced Vital Capacity (FVC), and FEV_1_/FVC. The three CIRS indices were able to predict the 12-month rate of moderate or severe exacerbation (CIRS Total Score: Hazard Ratio (HR) = 1.12 (95% CI: 1.08–1.21); CIRS Severity Index: HR = 1.21 (95% CI: 1.12–1.31); CIRS Comorbidity Index = 1.58 (95% CI: 1.33–1.89)). Conclusions. Among patients with COPD, the comorbidity number and severity, as assessed by the CIRS score, influence the risk in moderate-to-severe exacerbations. The CIRS score also correlates with the severity of respiratory symptoms and lung function.

## 1. Introduction

Chronic obstructive pulmonary disease (COPD) is the third leading cause of death worldwide, accounting for approximately 6% of total deaths worldwide according to the 2019 WHO report [1]. Clinical management of patients with high levels of complexity and multiple chronic conditions associated with COPD appears particularly challenging [2]. A meta-analysis by Putcha et al. showed that approximately 86% to 98% of COPD patients have at least one comorbidity (average number of comorbidities per individual = 1.2–4) [3]. The overlap between high levels of comorbidity and COPD is associated with a poor clinical and prognostic outcome: poorer quality of life and increased risk of exacerbations, hospitalization, and mortality [4,5,6,7]. This finding is similar to what has been found in studies of the role of comorbid conditions in other chronic respiratory diseases, such as asthma [8,9]. Although the underlying pathophysiological mechanisms linking COPD and comorbidities are not fully clarified, evidence suggests that a putative trait of union might be represented by the chronic low-grade systemic inflammatory status frequently observed in these patients [10]. Sarcopenia and subsequent muscle dysfunction may also play a role [11]. In this context, an important paradigm shift for patients may be a patient-centered approach with holistic and integrated management, rather than an approach focused on forced expiratory volume in the first second (FEV_1_)/dyspnea [12,13]. Strict adherence to recommendations or guidelines designed for the management of a single disease may be inappropriate for patients with multimorbidity [2], and there is a growing need for targeted tools for the management of highly complex COPD patients [14].

Major COPD staging systems often focus on respiratory dysfunction: modified BODE (mBODE), which includes body mass index, airflow obstruction, dyspnea, and exercise capacity; the ADO, composed of age, dyspnea, and airflow obstruction; or the modified DOSE (mDOSE), comprising dyspnea, airflow obstruction, smoking status, and exacerbation frequency [15]. In recent years, other multidimensional scales, such as the Charlson Comorbidity Index (CCI), the Comorbidity Test (COTE), and the Comorbidities in Chronic Obstructive Lung Disease (COMCOLD), have emerged to be predictors of mortality, hospitalization, exacerbation, and degree of dyspnea, providing physicians with new tools beyond lung function indices [16,17]. Their prognostic validation in COPD patients is well established [17]; however, they only assess the presence/absence of a disease and are poorly suited to explore the severity of the comorbidity burden. The Cumulative Illness Rating Scale (CIRS) is a well-validated multidimensional test commonly used as part of the Comprehensive Geriatric Assessment. CIRS score was originally developed by Linn et al. and offers a comprehensive disease assessment for 14 organ systems, based on a rating scale ranging from 0 to 4 [18]. The scale was later revised by Salvi F et al., who standardized the scoring system through concrete examples listed in the CIRS-G manual [19].

The main objective of this longitudinal prospective study was to determine the effectiveness of the CIRS score in predicting the risk of acute exacerbations in COPD patients; we hypothesized that a high severity index (CIRS-SI) and a high comorbidity index (CIRS-CI) would be associated with higher clinical severity of COPD, worse spirometric lung function, and/or higher incidence of acute exacerbations of COPD (AECOPD).

## 2. Materials and Methods

We recruited 105 participants with COPD who were referred to the Internal Medicine and Stroke Care Unit and Cardiovascular Risk Outpatient Unit of the Department of Promoting Health, Maternal–Infant. Excellence and Internal and Medicine (Promise) of the Policlinico Paolo Giaccone of the University of Palermo from 01/09/2021 to 01/09/2022. This ad interim analysis belongs to the ongoing MACH (Multidimensional Approach for COPD and High Complexity) Trial, registered on ClinicalTrial.gov Platform (NCT04986332), and was approved by Institutional review board (Comitato Etico Palermo 1; Approval Ref N. 04/2021). When completed, in 2026, the MACH study will have enrolled 300 subjects. The total follow-up period will be 36 months.

The primary objectives of the MACH Study will be the following:-To evaluate, in the COPD cohort, the impact of a multidimensional approach including global therapeutic remodeling (reassessment and therapeutic optimization of the total burden of chronic pathologic conditions) plus a 24-week moderate-intensity physical activity program on number of total and severe exacerbations, number of total and COPD/related hospitalizations, quality of life, and survival over a 36-month follow-up period;-To determine whether a higher index of comorbidity and polypharmacy are independently associated with worse clinical severity of COPD, increased risk of exacerbations, and reduced survival;-To determine the role of heart failure as the most significant comorbidity of COPD: assess the incidence rate and relative risk of total/severe exacerbations and the incidence rate and relative risk of total/COPD-related hospitalizations in heart-failure/COPD patients over 36 months of follow-up;

To be eligible to participate in this pilot study, an individual had to meet all the following criteria:Provide a signed and dated informed consent form;Be available for the duration of the study;Male or female aged >18 years;Ascertained COPD diagnosis according to the “Global Initiative for Chronic Obstructive Lung Disease” [14] or subjects who had never performed a spirometry with risk factors such as history or active tobacco smoke; occupational dust; vapor, fumes, gases, and other chemicals; and clinical indicators, such as chronic dyspnea and/or cough, recurrent wheezing, and shortness of breath upon exertion;History of ≥2 moderate exacerbations or ≥1 leading to hospitalization.Exclusion criteria were the following:Solid or hematological neoplasia under active (at the time of enrolment) or recent (completed less than 6 months earlier) chemoradiotherapy treatment at the time of enrolment;Pregnancy;Ongoing SARS-CoV-2 infection.

Each subject included in this analysis underwent a comprehensive medical history search, with particular attention paid to the assessment of COPD and comorbidities, pharmacological history, complete physical examination, and assessment of body mass index (BMI), calculated as the individual’s body weight divided by the square of height.

The present analysis was carried out after 12 months of follow-up, as follows: a reassessment was performed after 3, 6, and 12 months of discharge, referring the patients to the COPD and Cardiovascular Risk Outpatient Unit; each follow-up visit included a comprehensive clinical and therapeutic reassessment of the patients. During the control visits, information was collected on both moderate and severe acute exacerbations of COPD leading to hospitalization.

### 2.1. COPD Evaluation and Outcomes

The diagnosis of COPD was made according to the current GOLD report “Global Strategy for Prevention, Diagnosis and Management of COPD” [20]. For participants with previously diagnosed COPD, only spirometry tests performed within six months from enrolment were collected. For participants who met all inclusion criteria and had never performed a pulmonary function test, spirometry was performed in the outpatient clinic as soon as they were considered clinically stable and free from a respiratory tract infection. Spirometric measurements were performed using the POXY FX desktop spirometer (COSMED Srl, Rome, Italy). The procedures for Forced Vital Capacity (FVC) maneuvers were performed according to the statement “Standardization of Spirometry 2019 Update of American Thoracic Society/European Respiratory Society” [21], and for the comparison of measured values to the healthy population used the Global Lung Initiative (GLI) reference equation for spirometry [22]. Assessments of symptoms through the Modified British Medical Research Council (mMRC) and COPD Assessment Test (CAT™) were performed under stable COPD conditions. The primary outcome of the study was a composite of moderate or severe COPD exacerbation during the 12-month follow-up.

### 2.2. Administration of CIRS

One specifically trained research assistant administered the Cumulative Illness Rating Scale (CIRS). Three indices were derived from the CIRS: the total score (CIRS-TS) or the total scores of the 14 system scores; the severity index (CIRS-SI) or mean of the scores of the first 13 categories (excluding the category of psychiatric/behavioral pathologies); and the comorbidity index (CIRS-CI) or the number of categories in which a score greater than or equal to 3 is obtained (excluding the category of psychiatric/behavioral pathologies) [19]. The CIRS score is a valid indicator of health status among residents of frail older institutions [19,23].

The complete validated CIRS [19] administered is reported below: **Body System****Score**1. Cardiac (heart only)012342. Hypertension (rating is based on severity; organ damage is rated separately)012343. Vascular (blood, blood vessels and cells, bone marrow, spleen, and lymphatics)012344. Respiratory (lungs, bronchi, and trachea below the larynx)012345. EENT (eye, ear, nose, throat, and larynx)012346. Upper GI (esophagus, stomach, and duodenum; pancreas; does not include diabetes)012347. Lower GI (intestines and hernias)012348. Hepatic (liver and biliary tree)012349. Renal (kidneys only)0123410. Other GU (ureters, bladder, urethra, prostate, and genitals)0123411. Musculoskeletal–integumentary (muscle, bone, and skin)0123412. Neurological (brain, spinal cord, and nerves; does not include dementia)0123413. Endocrine–Metabolic (includes diabetes, thyroid; breast; systemic infections; and toxicity)0123414. Psychiatric/Behavioral (includes dementia, depression, anxiety, agitation/delirium, and psychosis)01234

### 2.3. Statistical Analysis

A statistical analysis of quantitative and qualitative data, including descriptive statistics, was performed for all data collected. Continuous variables were expressed as mean ± standard deviation (SD), and categorical variables were expressed as frequency counts and percentages. The CIRS indices were analyzed as mean ± SD and as tertiles of the distribution. The correlation analysis was performed using Spearman’s nonparametric test. The Cox proportional hazards regression was used to assess the ability of the three tertiles of CIRS indices to predict the primary outcome. Cox regression models were adjusted for confounder variables, age, sex, CAT score (<10 points as reference variable), and GOLD class (Class 1 as reference variable). A two-tailed *p*-value < 0.05 was considered significant, and a 95% confidence interval (CI) was reported. CIRS-TS and CIRS-SI showed collinearity, so only the latter was imputed to the regression models. A statistical analysis was performed using STATA Statistical Software, version 17 (Stata-Corp, College Station, TX-USA). For data visualization of the multivariate logistic regression models and the Cox regression models, we used GraphPad Prism 9.5.0 software (Graph Pad Software, San Diego, CA, USA).

## 3. Results

All 105 participants completed the study. Eight participants died during the 12 months of follow-up, three from cardiovascular complications and the remaining five from a lower respiratory tract infection complicated by respiratory failure.

### 3.1. Demographic, Anthropometric Variables, and Pattern of Comorbidities of Enrolled Participant

The baseline characteristics of the 105 study participants are presented in Table 1. The mean age is 72.13 ± 9.07 years, and 67.95% are men. Active smokers are 36.8%. Cardiovascular diseases are highly prevalent in this sample: 80% are hypertensive, and 73% are affected by heart failure (HF); interestingly, HF with preserved ejection fraction (HFpEF) is more represented (48.6%). A total of 30.5% of them have atrial fibrillation. There is a small group of subjects with asthma-COPD overlap syndrome (ACOS) (4.76%) and obstructive sleep apnea syndrome (OSAS)/COPD (8.57%); the small sample size does not allow for a comparative analysis at this time.

### 3.2. Spirometric Assessment of COPD Participants

The spirometric system used to assess disease severity is shown in Table 2. According to the inclusion criteria of “≥2 moderate exacerbations or ≥1 leading to hospitalization”, all COPD participants belonged to the “E” group, as recently proposed by the 2023 GOLD report [20].

### 3.3. Multidimensional Assessment

The multidimensional assessment is shown in Table 3.

### 3.4. Results of the Spearman Correlation Analysis

Table 4 shows the results of the Spearman correlation analysis. Our results clearly show that the CIRS indices correlate with the breathlessness severity assessed by the mMRC scale, with spirometric variables, specifically with the severity of airflow limitation based on the value of FEV_1_.

### 3.5. Cox Regression Analysis

The most significant results of the Cox regression analysis are displayed in Figure 1 and Figure 2. The primary composite outcome occurred in 52 participants (49.52%). The first multivariate Cox regression models were computed, accounting for age, gender, CAT > 10 points, and GOLD Class as major confounding variables; they were computed by entering CIRS indices as continuous variables. We found that a one-point increase in CIRS-TS, CIRS-SI, and CIRS-CI was associated with a 12%, 16%, and 58% higher risk of acute exacerbation, respectively (see Table 5 and Figure 1). We also performed a multivariate Cox regression, entering the CIRS indices individually as tertiles. The results confirm that, in our cohort, a higher illness severity is associated with a higher risk of exacerbation during the 12-month follow-up (see Table 5).

Finally, when the multivariate Cox regression models were computed by entering both high and low tertile of CIRS indices, the risk was higher, with a greater amount of disease severity and comorbidities (see Table 6 and Figure 2).

## 4. Discussion

The main results of our analysis would support the hypothesis that CIRS scoring—thus, using a validated tool of a multidimensional assessment—is suitable for predicting COPD outcomes in a population of COPD subjects with a high level of clinical complexity and high frequency of exacerbations. The CIRS score appears to be a reliable test to assess the overall status of COPD subjects and correlates significantly with the respiratory symptoms assessed through the mMRC (*p* < 0.001) and with the severity of airflow limitation (*p* = 0.002). The multivariate Cox regression analysis showed that the risk of the primary endpoint (moderate or severe AECOPD) increased in a directly proportional manner to the value of the two CIRS-derived scores (SI and CI).

Our findings are supported by the current literature: the presence of comorbidities has a major impact on COPD outcomes, and having a significant burden of comorbidities is associated with an increased risk of exacerbation [3,24]. However, to the best of our knowledge, this is the first attempt to address the burden of comorbidity with a multidimensional tool that is not specifically tailored to COPD patients and able to assess not only the number of diseases but also their severity, with promising results.

The correlation analysis confirms that the CIRS indices provide the clinician with a comprehensive and reliable view of the complexity of COPD patients: they correlate with the degree of airflow limitation; with the degree of independence, as assessed by the Barthel Index; with polytherapy; and with the severity of breathlessness reported by patients, according to the mMRC and CAT, thus supporting the hypothesis that not only the burden of non-respiratory comorbidities may influence the severity of respiratory symptoms, but also that the risk of exacerbations may be related and effectively estimated through the analysis of non-respiratory variables [25,26,27].

Current management strategies for patients with comorbid COPD are not clearly established worldwide, but a personalized and integrated approach that goes beyond FEV_1_ and includes both COPD severity and the burden of comorbidities is urgently needed [28]. Given this, CIRS scoring in a high-clinical-complexity setting appears to be a reliable tool to obtain a comprehensive evaluation of the global burden of comorbidity and illness severity of this category of subjects, able to effectively represent multiple clinical aspects of COPD subjects, as demonstrated by the numerous correlations shown in the ongoing MACH Study with other more targeted tests to assess disability (Barthel Index), nutritional status (Mini Nutritional Assessment), and quality of life (EQ-5D-3L).

Based on the Cox regression results, CIRS scoring shows consistent performance in predicting new acute exacerbations over a relatively short follow-up period of 52 weeks. Our results confirm the association between comorbidities and an increased risk of exacerbations [3,4,24,25,29]. The mechanisms underlying this association are still the subject of investigation, and there is no consensus whether comorbidities directly provoke exacerbations, mimicking exacerbation symptoms or even overlapping them [30]. However, the ability of CIRS to predict the risk of a new severe exacerbation in high-complexity COPD patients, particularly in the first months after hospitalization, suggests a tailored approach in which the management of acute exacerbations in high-complexity COPD patients could be planned with closer monitoring of clinical status, adherence to therapy, and degree of dyspnea in the first months immediately after hospital discharge. In this sense, CIRS could be a new, helpful tool for physician decision making in both short- and long-term management. Other new approaches are being developed in this area, such as the omics approach to finding new blood biomarkers using sequencing techniques and multiplex platforms that can measure several thousand gene transcripts, proteins, or metabolites [31] or the search for microRNA (miRNA). The deregulation of miRNAs targeting a variety of cellular and molecular pathways may be involved in COPD pathogenesis; the identification of particular miRNAs associated with specific COPD phenotypes may provide interesting biomarkers of disease severity and prognosis [32].

Our study has several limitations. The main limitation is the limited number of subjects enrolled. This study is intended to be a pilot analysis of the ongoing MACH study. Given the relatively small sample size, the conclusions are preliminary: our results deserve an in-depth analysis and validation by enrolling a larger number of subjects, but several interesting findings emerged already after a short follow-up of 12 months in 105 patients. Second, our approach is appropriate for a population of COPD patients with high clinical complexity, high risk of exacerbations, and a narrow range of patient ages, so we cannot extrapolate our results to all COPD patients, especially those with less complicated disease and lower overall risk. This is a limitation, but one of the main aims of the MACH study is to provide information on such a group of patients who are not usually included in clinical trials due to their high complexity and poor prognosis. Therefore, we believe that our results for this specific target population are relevant. Thirdly, a comparison with other multidimensional tests already validated in COPD patients, such as the Body Mass Index, Airflow Obstruction, Dyspnea, and Exercise Capacity (BODE) index; COMCOLD; CCI; and COTE, is needed to demonstrate the actual better prognostic role of CIRS in this specific patient population. Fourthly, in our analysis, the medications taken by the patients (not only for the respiratory disease) seem to have a relevant prognostic role (both positive and negative), but this is an aspect that cannot be investigated using the CIRS score. This is a key issue because of the extremely close relationship between the severity of chronic disease and the adequacy of treatment. Finally, the use of CIRS is subject to the judgement of the operator, and the usual area of application is geriatrics. Therefore, the real potential of our approach in a context other than the one proposed could not be established.

## 5. Conclusions

COPD is a heterogeneous systemic syndrome rather than a respiratory disease [8] and encompasses several clinical entities that occur during the natural history of the disease. The identification of comorbidities in COPD patients appears to be crucial because of their relevant prognostic value and since a significant percentage of COPD deaths can be attributed to their comorbidities rather than to COPD itself [33]. From this perspective, CIRS scoring could therefore provide a useful screening protocol for comorbidities, while identifying COPD patients with a greater need for clinical monitoring, thus stratifying highly complex COPD patients and optimizing the number and frequency of periodic follow-up visits. Several authors emphasize the importance of a multidisciplinary approach to the management of COPD, especially when multiple comorbidities coexist; these should guide follow-up strategies, as they are directly related to the clinical severity and frequency of exacerbations [34,35].

In conclusion, our study highlights the potentially relevant role of a multidimensional approach for highly complex comorbid COPD patients. The CIRS score could be a useful tool not only to provide an accurate measure of the burden of comorbidity and disease severity in such a group of patients but also to predict the occurrence of new exacerbations; our experience could be useful for the development of a new model of targeted follow-up with an integrated personalized care that also takes into account comorbidities [36].

## Figures and Tables

**Figure 1 jpm-13-01674-f001:**
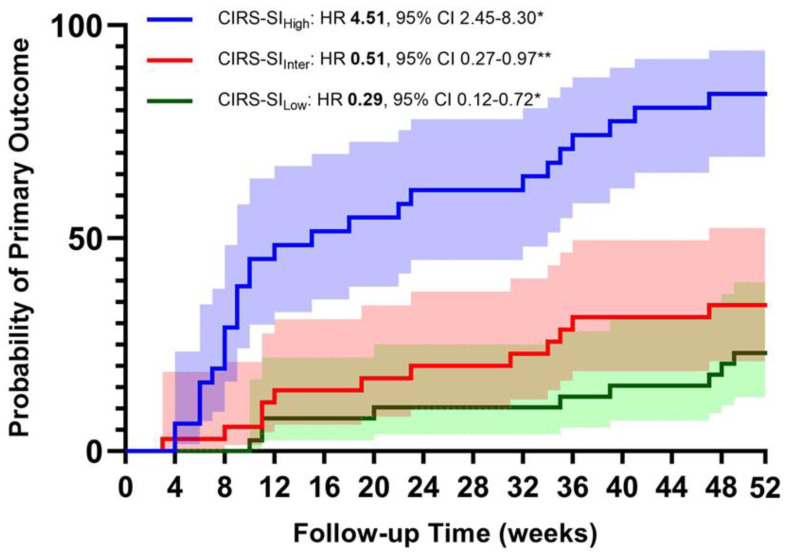
Probability of moderate to severe exacerbations according to the tertile of CIRS Severity Index (* = *p* < 0.01; ** = *p* < 0.05).

**Figure 2 jpm-13-01674-f002:**
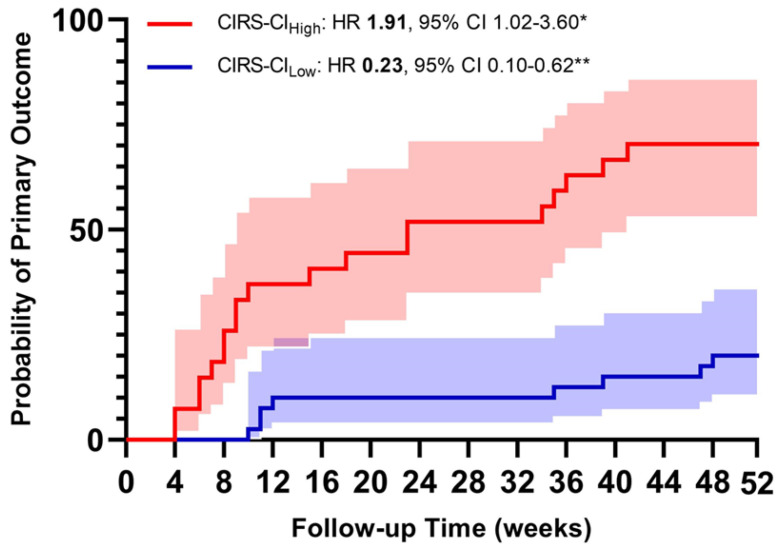
Probability of moderate-to-severe exacerbations according to the highest vs. the lowest tertile of the CIRS Comorbidity Index (* = *p* < 0.05; ** = *p* < 0.01).

**Table 1 jpm-13-01674-t001:** Demographic and anamnestic variables of COPD patients at admission.

Variable		Count (%)	Mean ± SD
Male, *n* (%)		69 (67.95)	
Age (yrs), mean ± SD			72.13 ± 9.07
BMI (Kg/m^2^), mean ± SD			28.59 ± 6.07
Former smoker (yes), *n* (%)		54 (50.94)	
Active smoker (yes), *n* (%)		39 (36.79)	
Environmental risk factors (yes), *n* (%)		33 (31.73)	
Hypertension (yes), *n* (%)		84 (80)	
Heart failure (yes), *n* (%)		72 (73.08)	
	HFpEF (yes), *n* (%)	52 (48.57)	
	HFmrEF (yes), *n* (%)	6 (7.62)	
	HFrEF (yes), *n* (%)	14 (16.89)	
Ejection fraction (%), mean ± SD			52.34 ± 8.22
Atrial fibrillation (yes), *n* (%)		32 (30.48)	
Chronic kidney disease (yes), *n* (%)		54 (51.43)	
Peripheral artery disease (yes), *n* (%)		15 (14.29)	
Chronic gastritis/history of gastric/duodenal ulcer (yes), *n* (%)		13 (12.38)	
Previous DTV/PE (yes), *n* (%)		8 (7.62)	
TIA/Ischemic Stroke (yes), *n* (%)		14 (13.33)	
Chronic coronary syndrome (yes), *n* (%)		35 (33.33)	
Cirrhosis (yes), *n* (%)		3 (2.86)	
Dyslipidemia (yes), *n* (%)		75 (71.43)	
Asthma (yes), *n* (%)		5 (4.76)	
Type 2 diabetes mellitus (yes), *n* (%)		46 (43.81)	
Hypothyroidism (yes), *n* (%)		17 (16.19)	
Hyperthyroidism (yes), *n* (%)		9 (8.57)	
Obstructive sleep apnea syndrome (yes), *n* (%)		9 (8.57)	
Chronic respiratory failure (yes), *n* (%)		37 (34.91)	
Prostatic hypertrophy (yes), *n* (%)		16 (23.19)	
Total number of active substances, mean ± SD			9.25 ± 3.57
Insulin (yes), *n* (%)		25 (23.81)	
Metformin (yes), *n* (%)		9 (8.57)	
Gliflozin (yes), *n* (%)		13 (12.38)	
GLP1 agonist/DPP-4 inhibitors (yes), *n* (%)		7 (6.67)	
Antiplatelet (yes), *n* (%)		39 (37.5)	
Antithrombotic therapy (yes), *n* (%)		32 (30.48)	
PPI (yes), *n* (%)		81 (77.14)	
Statin (yes), *n* (%)		59 (56.19)	
Erythropoietin (yes), *n* (%)		6 (5.71)	
Levothyroxine (yes), *n* (%)		13 (12.38)	
Diuretic therapy (yes), *n* (%)		70 (66.67)	
β-blocker therapy (yes), *n* (%)		56 (53.33)	
ACEi/ARB (yes), *n* (%)		61 (58.1)	
Calcium channel blockers (yes), *n* (%)		34 (32.38)	

Data are presented as mean value ± SD; BMI: Body Mass Index; HFpEF: heart failure with preserved ejection fraction; HFmrEF: heart failure with mildly reduced ejection fraction; HFrEF: heart failure with reduced ejection fraction; DTV: deep vein thrombosis; PE: pulmonary embolism; TIA: transient ischemic attack; PPI: proton pump inhibitor; GLP1: glucagon peptide-like 1; DPP-4: dipeptidyl peptidase-4; ACEi: angiotensin-converting enzyme inhibitors; ARB: angiotensin II receptor blocker.

**Table 2 jpm-13-01674-t002:** COPD-related variables of the enrolled participant.

Variable		Count (%)	Mean ± SD
mMRC, *n* (%)	0	5 (4.76)	
	1	19 (18.1)	
	2	28 (26.67)	
	3	35 (33.33)	
	4	18 (17.14)	
CAT, mean ± SD			16.45 ± 7.64
FVC (Lt), mean ± SD			2.31 ± 0.7
FVC (% predicted), mean ± SD			73.64 ± 17.27
FEV_1_ (Lt/s), mean ± SD			1.44 ± 0.55
FEV_1_ (% predicted), mean ± SD			58.55 ± 18.31
FEV_1_/FVC, mean ± SD			0.61 ± 0.08
COPD-GOLD class, *n* (%)	GOLD 1E	12 (11.43)	
	GOLD 2E	54 (51.43)	
	GOLD 3E	35 (33.33)	
	GOLD 4E	4 (3.81)	
Inhaled bronchodilators, *n* (%)	LAMA	21 (20)	
	LABA + ICS	18 (17.14)	
	LABA + LAMA	32 (30.48)	
	LABA + LAMA + ICS	34 (32.38)	

Data are presented as mean value ± SD; mMRC: modified Medical Research Council; CAT: COPD Assessment Test; FVC: Forced Vital Capacity; FEV_1_: forced expiratory volume in the 1st second; COPD: chronic obstructive pulmonary disease; GOLD: Global Initiative for Chronic Obstructive Lung Disease; LAMA: long-acting muscarinic antagonist; LABA: long-acting β-agonist; ICS: inhaled corticosteroid.

**Table 3 jpm-13-01674-t003:** Multidimensional test score.

Mean ± SD		Tertile Low	Tertile Intermediate	Tertile High
		*n*, Mean ± SD	*n*, Mean ± SD	*n*, Mean ± SD
CIRS-TS	17.72 ± 6.58	39, 10.89 ± 3.15	35, 18.48 ± 1.93	31, 25.45 ± 3.23
CIRS-SI	1.34 ± 0.48	39, 0.82 ± 0.24	35, 1.41 ± 0.14	31, 1.91 ± 0.19
CIRS-CI	3.13 ± 1.76	40, 1.37 ± 0.66	38, 3.26 ± 0	27, 5.55 ± 0.80

CIRS-TS: Cumulative Illness Rating Scale–Total Score; CIRS-SI: Cumulative Illness Rating Scale—Severity Index; CIRS-CI: Cumulative Illness Rating Scale—Comorbidity Index.

**Table 4 jpm-13-01674-t004:** Spearman’s correlation analysis.

		mMRC	CAT	FVC (Lt)	FVC (%)	FEV_1_ (Lt)	FEV_1_ (%)	CIRS-TS	CIRS-IS	CIRS-IC
CIRS-TS	ρ	0.497 *	0.315 *	−0.263 **	−0.318 *	−0.256 **	−0.299 **	--	0.993 *	0.892 *
CIRS-SI	ρ	0.504 *	0.317 *	−0.268 **	−0.327 *	−0.260 **	−0.298 **	0.991 *	--	0.897 *
CIRS-CI	ρ	0.498 *	0.329 *	−0.269 **	−0.331 *	−0.273 **	−0.313 **	0.876 *	0.897 *	--

CIRS-TS: Cumulative Illness Rating Scale—Total Score; CIRS-SI: Cumulative Illness Rating Scale—Severity Index; CIRS-CI: Cumulative Illness Rating Scale—Comorbidity Index; mMRC: modified Medical Research Council; FEV_1_: forced expiratory volume in the 1st second; * *p* < 0.001; ** *p* <0.01.

**Table 5 jpm-13-01674-t005:** Results of Cox regression analysis according CIRS indices as continuous variables and as tertiles.

		Hazard Ratio	SE		*p*-Value	95% Confidence Interval	
CIRS-TS		1.15	0.016		<0.001	1.08	1.21
CIRS-SI		1.21	0.049		<0.001	1.12	1.31
CIRS-CI		1.58	0.141		<0.001	1.33	1.89
CIRS-SI Tertile	High	4.51	1.4	3.93	<0.001	2.45	8.30
	Intermediate	0.51	0.16	−0.28	0.04	0.27	0.97
	Low	0.29	0.13	−2.89	0.007	0.12	0.72
CIRS-CI Tertile	High	2.68	0.84	3.41	0.002	1.45	4.96
	Intermediate	1.05	0.31	1.23	0.861	0.58	1.89
	Low	0.21	0.01	−3.80	0.001	0.1	0.52

CIRS-TS: Cumulative Illness Rating Scale—Total Score; CIRS-SI: Cumulative Illness Rating Scale—Severity Index; CIRS-CI: Cumulative Illness Rating Scale—Comorbidity Index; SE: standard error.

**Table 6 jpm-13-01674-t006:** Multivariate Cox regression model: low vs. high tertile of Severity Index and Comorbidity Index.

		Hazard Ratio	SE	*p*-Value	95% Confidence Interval	
CIRS-SI Tertile	High	3.90	1.32	<0.001	2.00	7.60
	Low	0.48	0.24	0.150	0.18	1.29
CIRS-CI Tertile	High	1.91	0.61	0.04	1.02	3.60
	Low	0.23	0.11	0.003	0.10	0.62

CIRS-SI: Cumulative Illness Rating Scale—Severity Index; CIRS-CI: Cumulative Illness Rating Scale—Comorbidity Index; SE: standard error.

## Data Availability

The archive of the research data is fully accessible by requesting it directly to the authors of the MACH study from which this pilot analysis was derived.

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
