# Peer review of "The Role of the Cumulative Illness Rating Scale (CIRS) in Estimating the Impact of Comorbidities on Chronic Obstructive Pulmonary Disease (COPD) Outcomes: A Pilot Study of the MACH (Multidimensional Approach for COPD and High Complexity) Study"

_jpm, 2023, doi:10.3390/jpm13121674_

Round 1

Reviewer 1 Report

Comments and Suggestions for Authors

Raimondo et al. studied the potential as a predictor of acute exacerbation by scoring the assessment of comorbidities with CIRS indices in COPD patients. COPD can affect multiple systemic diseases, and concomitant diseases can also have some prognostic significance through interaction with COPD, laying the foundation for an integrated and individualized approach to COPD patients. If authors present the CIRC index in a table, it will be able to help readers understand and use them as reference materials in future research and clinical practice. In addition, for the first abbreviation, please present the full name from the Abstract through the manuscript. It would be more interesting if authors could suggest the difference in the study results between COPD alone and COPD-asthma overlap syndrome. Although it is a paper of public importance, please specify the name of the pilot study in the title because the number of participants is small, and the results of the nonparametric test are included.

Comments on the Quality of English Language

The quality of the overall English language is good, which is judged suitable for readers worldwide to understand. For the first abbreviation, please suggest the full name from the Abstract.

Author Response

REVIEWER #1

We would like to thank you for the detailed review of our manuscript. We greatly appreciate the effort you made concerning your critique for the review of our study. We have accepted all your suggestions and revised the article according to them.

Di Raimondo et al. studied the potential as a predictor of acute exacerbation by scoring the assessment of comorbidities with CIRS indices in COPD patients. COPD can affect multiple systemic diseases, and concomitant diseases can also have some prognostic significance through interaction with COPD, laying the foundation for an integrated and individualized approach to COPD patients.

If authors present the CIRC index in a table, it will be able to help readers understand and use them as reference materials in future research and clinical practice.

Thank you for your suggestion, we have included the CIRS score calculation as a table in the revised version of our manuscript.

In addition, for the first abbreviation, please present the full name from the Abstract through the manuscript.

Thank you for your valuable comment: we have added the full names of all achronyms as they first appeared, both in the abstract and in the manuscript.

It would be more interesting if authors could suggest the difference in the study results between COPD alone and COPD-asthma overlap syndrome.

That's an interesting point, given several reports that seem to indicate a better prognosis for asthma-COPD overlap syndrome (ACOS) subjects vs COPD alone. IN this pilot analysis there is a small group of subjects with ACOS (4.76%) and obstructive sleep apnea syndrome (OSAS)/COPD (8.57), but the small sample size does not allow a comparative analysis of statistical significance at this time. We have added this sentence in the revised version of the manuscript.

Although it is a paper of public importance, please specify the name of the pilot study in the title because the number of participants is small, and the results of the nonparametric test are included.

Thank you for your comment: we have changed the title of the manuscript in accordance with your and reviewer #2's suggestions.

The quality of the overall English language is good, which is judged suitable for readers worldwide to understand. For the first abbreviation, please suggest the full name from the Abstract.

Done.

We hope that we have successfully changed our manuscript according to your suggestions and that we have provided all the necessary explanations. We also hope that the manuscript now fulfills your criteria, and the Journal criteria for publication.

Reviewer 2 Report

Comments and Suggestions for Authors

The manuscript “Influence of comorbidities on chronic obstructive pulmonary disease (COPD) outcomes: risk prediction of acute exacerbations using the cumulative illness rating scale (CIRS)” deals with the determination of the effectiveness of the CIRS score in detecting the association of comorbidities and disease severity with the risk of acute exacerbations in COPD patients.

The topic discussed is very important in the treatment COPD patients. CIRS scoring could therefore provide a useful screening protocol for comorbidities and will allow you to implement a personalized approach to COPD patients.

I would like to make a few comments:

1)      The title is too long; perhaps it is better to shorten it.

2)      Line 19: Add please the age of the patients.

3)      In the Abstract add please the explanation of abbreviations mMRC  and CAT.

4)      Lines 119-122: “CIRS score was originally developed by Linn et al and offers a comprehensive disease assessment for 14 organ systems, based on a rating scale ranging from 0 to 4 [17]. The scale was later revised by Miller et al. which standardize the scoring system through concrete examples listed in the CIRS-G manual [18]”.

Comment: perhaps it is better to replace this information to the Introduction.

5)      In the Discussion, it is necessary to mention omics approaches aimed to determine the markers of phenotype, severity and associated pathologies. The use of omics approach markers will help expand the ability to predict complications in COPD patients.

6)      One more limitation of the study should be mentioned: narrow range of patient ages.

Author Response

REVIEWER #2:

We would like to thank you for your expert review of our manuscript. Thank you very much for your positive opinion regarding our manuscript. We put a lot effort in this study and we appreciate your opinion very much.

The manuscript “Influence of comorbidities on chronic obstructive pulmonary disease (COPD) outcomes: risk prediction of acute exacerbations using the cumulative illness rating scale (CIRS)” deals with the determination of the effectiveness of the CIRS score in detecting the association of comorbidities and disease severity with the risk of acute exacerbations in COPD patients.

The topic discussed is very important in the treatment COPD patients. CIRS scoring could therefore provide a useful screening protocol for comorbidities and will allow you to implement a personalized approach to COPD patients.

I would like to make a few comments:

1)      The title is too long; perhaps it is better to shorten it.

Thank you for your comment: we have changed the title of the manuscript in accordance with your and reviewer #2's suggestions. Unfortunately, reviewer #1 asked us to add a reference in the title to the MACH study, of which this article is a pilot study. The title is not shorter, but I hope it is clearer and more complete.

2)      Line 19: Add please the age of the patients.

Thank you for your suggestion, we have added this information in the abstract.

3)      In the Abstract add please the explanation of abbreviations mMRC and CAT.

Thank you for your comment: we have added the full names of all acronyms as they first appeared, both in the abstract and in the manuscript.

4)      Lines 119-122: “CIRS score was originally developed by Linn et al and offers a comprehensive disease assessment for 14 organ systems, based on a rating scale ranging from 0 to 4 [17]. The scale was later revised by Miller et al. which standardize the scoring system through concrete examples listed in the CIRS-G manual [18]”. Comment: perhaps it is better to replace this information to the Introduction.

Thank you for the good suggestion. We have moved this information to the introduction section.

5)      In the Discussion, it is necessary to mention omics approaches aimed to determine the markers of phenotype, severity and associated pathologies. The use of omics approach markers will help expand the ability to predict complications in COPD patients.

This is a very interesting topic. We have added a brief consideration of this aspect of the future management of COPD in the Discussion section.

6)      One more limitation of the study should be mentioned: narrow range of patient ages.

Thank you for the suggestion, we added this further limitation.

We hope that we have successfully changed our manuscript according to your suggestions and that we have provided all the necessary explanations. We also hope that the manuscript now fulfills your criteria, and the Journal criteria for publication.